# Di-(2-ethylhexyl) Phthalate Triggers Proliferation, Migration, Stemness, and Epithelial–Mesenchymal Transition in Human Endometrial and Endometriotic Epithelial Cells via the Transforming Growth Factor-β/Smad Signaling Pathway

**DOI:** 10.3390/ijms23073938

**Published:** 2022-04-01

**Authors:** Hwi Gon Kim, Ye Seon Lim, Seonyeong Hwang, Hye-Yoon Kim, Yuseok Moon, Yong Jung Song, Yong-Jin Na, Sik Yoon

**Affiliations:** 1Department of Obstetrics and Gynecology, Pusan National University College of Medicine, Yangsan 626-870, Gyeongsangnam-do, Korea; bislsan@naver.com (H.G.K.); gynsong@gmail.com (Y.J.S.); yjna@pusan.ac.kr (Y.-J.N.); 2Department of Anatomy and Convergence Medical Sciences, Pusan National University College of Medicine, Yangsan 626-870, Gyeongsangnam-do, Korea; yeseonlim@pusan.ac.kr (Y.S.L.); anatomy2017@pusan.ac.kr (S.H.); solarhy77@naver.com (H.-Y.K.); 3Immune Reconstitution Research Center of Medical Research Institute, Pusan National University College of Medicine, Yangsan 626-870, Gyeongsangnam-do, Korea; moon@pusan.ac.kr; 4Department of Convergence Medical Sciences, Pusan National University College of Medicine, Yangsan 626-870, Gyeongsangnam-do, Korea

**Keywords:** DEHP, endometriosis, inflammation, epithelial–mesenchymal transition, stem cell, transforming growth factor-β

## Abstract

Di-(2-ethylhexyl) phthalate (DEHP) is a frequently used plasticizer that may be linked to the development of endometriosis, a common gynecological disorder with a profound impact on quality of life. Despite its prevalence, vital access to treatment has often been hampered by a lack of understanding of its pathogenesis as well as reliable disease models. Recently, epithelial–mesenchymal transition (EMT) has been suggested to have a significant role in endometriosis pathophysiology. In this study, we found that DEHP treatment enhanced proliferation, migration, and inflammatory responses, along with EMT and stemness induction in human endometrial and endometriotic cells. The selective transforming growth factor-β (TGF-β) receptor type 1/2 inhibitor LY2109761 reversed the DEHP-induced cell proliferation and migration enhancement as well as the increased expression of crucial molecules involved in inflammation, EMT, and stemness, indicating that DEHP-triggered phenomena occur via the TGF-β/Smad signaling pathway. Our study clearly defines the role of DEHP in the etiology and pathophysiological mechanisms of endometriosis and establishes an efficient disease model for endometriosis using a biomimetic 3D cell culture technique. Altogether, our data provide novel etiological and mechanistic insights into the role of DEHP in endometriosis pathogenesis, opening avenues for developing novel preventive and therapeutic strategies for endometriosis.

## 1. Introduction

Globally, plastic production and consumption have been rapidly increasing, and it is nearly impossible to imagine our modern world without plasticware. Plastic manufacturers continued to increase production even during the COVID-19 pandemic when most industries suffered the consequences of a global economic crisis. Although plasticware is necessary in major industries that help to improve the quality of our lives, plastic pollution has been simultaneously and increasingly recognized as a serious threat to human health and ecosystems.

Several harmful substances are associated with plastics, such as phthalate esters (PAEs), which are one of the most prevalent persistent organic pollutants in the environment; they are colorless, odorless, synthetic oily compounds with low volatility and water solubility [1]. PAEs are widely used as plasticizers to impart flexibility, pliability, and elasticity in polymer products, softeners in polyvinyl chloride plastics, and fragrance stabilizers in hygiene and cosmetic products [2]. In particular, di-(2-ethylhexyl) phthalate (DEHP) is one of the most frequently used PAEs in several consumer products such as food and beverage packages, soft squeeze toys, building and construction materials, household goods, personal care products, film coatings for pharmaceutical tablets, and medical devices and equipment [2,3].

DEHP and its metabolites are generally regarded as endocrine-disrupting chemicals (EDCs), which are defined as a class of exogenous chemicals that alter the function of the endocrine system and have adverse effects on reproductive, developmental, cardiovascular, skeletal, neurological, metabolic, and immune processes [4]. EDCs primarily target reproductive organs, and DEHP induces multiple developmental and reproductive toxicities; in particular, it is significantly associated with the occurrence of some important gynecological diseases such as endometriosis [5,6]. However, the precise etiology, pathogenesis, and molecular targets of DEHP-associated endometriosis remain elusive. DEHP also affects transforming growth factor-β1 (TGF-β1) signaling, thus causing hypospadias [7]. Moreover, DEHP and its metabolites promote hepatic fibrosis and enhance the migratory and invasive potential of human melanoma cells via TGF-β1 signaling activation [8,9].

Endometriosis is a common, chronic, often debilitating, and benign gynecological disorder characterized by the presence of ectopic endometrial glands and stroma outside the uterus [10]. In particular, it is closely associated with inflammatory reactions and the resultant fibrosis, leading to typical symptoms and complications including chronic and often intractable pelvic pain, dysmenorrhea, dyspareunia, and infertility [11]. Peritoneal adhesion and outgrowth of endometrial cells due to retrograde menstruation are believed to be major factors in endometriosis pathogenesis [10]. In addition to the classic implantation hypothesis, alternative hypotheses such as metaplasia [12], inflammatory and immunologic responses [11], peritoneal barrier disturbance [13], genetic and epigenetic changes [14], circulating stem/progenitor cells [15], and repeated tissue injury and repair [16] have also been suggested. Moreover, the altered intestinal absorption can cause transposition of intestinal microbiota and the inflammation of the peritoneal fluid [17], highlighting the relationships between endometriosis and the host microbiome [18]. However, the stimuli promoting the attachment and outgrowth of endometrial cells in the peritoneal cavity remain largely unknown.

Among many factors proposed to be involved in endometriosis pathogenesis, epithelial–mesenchymal transition (EMT) has recently been found to play a crucial role in endometriosis development and progression [19,20]. EMT is a complex cellular process by which epithelial cells lose polarity and cell–cell adhesion, acquiring migratory and invasive properties for transition to a mesenchymal state, and gaining stem-like features [21]. Once a cell becomes a stem cell, it develops stem cell-specific properties such as self-renewal, stemness maintenance, and the ability to differentiate into multiple cell types. For instance, the transition of cancer cells into cancer stem cells through EMT activation has been proposed as a key step in metastasis, chemoresistance, and recurrence [21]. Hence, EMT plays essential roles in specific steps of embryogenesis, tissue morphogenesis, and organogenesis during development, as well as wound healing, tissue regeneration, inflammation, and fibrosis [22,23].

In multicellular organisms, the maintenance of highly organized, dynamic, and structurally integrated systems critically relies on the establishment of intercellular cohesiveness and communication as well as the control and coordination of cellular activities; this is brought about by various chemical and mechanical signals regulating fundamental physiological processes such as proliferation, apoptosis, differentiation, and metabolism, allowing for the specialization of groups of cells [24]. Among the many signaling pathways, the TGF-β pathway plays a central role in inducing EMT in different tissue types [25,26].

A need to improve the understanding of endometriosis progression and treatment has driven the development of accurate and physiologically relevant in vitro 3D cell culture models, which serve as important tools in endometriosis research [27,28,29].

This study shows that DEHP plays an important role in the etiology and pathogenesis of endometriosis by enhancing cell proliferation and migration and inducing EMT and stem cell properties via the TGF-β/Smad signaling pathway in human endometrial and endometriotic epithelial cells (EEECs) using a biomimetic hydrogel-based 3D culture model. Furthermore, our findings suggest that the TGF-β/Smad signaling pathway is a potential therapeutic target for DEHP-induced endometriosis.

## 2. Results

### 2.1. DEHP Promotes Human EEECs Proliferation

A CCK-8 cell proliferation assay was performed after treatment with different DEHP concentrations to assess the effects of DEHP treatment on human EEEC proliferation, which play an important role in endometriosis development. The proliferation rates of Ishikawa, End1/E6E7, and VK2/E6E7 cells were enhanced 1.2-fold, 1.2-fold, and 1.3-fold (all *p* < 0.01), respectively, after incubation with 25 µM DEHP for 48 h (Figure 1A).

Further, these results were consistent with the morphological and immunocytochemical characteristics observed using phase contrast microscopy (Figure 1B), as well as staining of the cell cycle-associated nuclear antigen Ki-67, which is a cellular marker for cell proliferation (Figure 1C). Thus, these results strongly suggest that DEHP triggers human EEEC proliferation.

### 2.2. DEHP Enhances Human Endometrial Epithelial Cell Proliferation through TGF-β Signaling Pathway

The level of phosphorylated Smad-2 (pSmad-2), which represents canonical TGF-β signaling pathway activation, was assessed using a Western blot assay after treatment with several DEHP concentrations for 24 h to determine whether the effects of DEHP were mediated by an increased response to the TGF-β signaling pathway in human endometrial epithelial cells. pSmad-2 level in Ishikawa cells after incubation with 5, 10, and 25 µM DEHP significantly increased 1.3-fold (*p* < 0.001), 1.4-fold (*p* < 0.01), and 1.6-fold (*p* < 0.01), respectively, in a dose-dependent manner (Figure 2A).

Next, we evaluated LY2109761 toxicity on human endometrial epithelial cells using a CCK-8 cell proliferation assay to validate TGF-β signaling inhibition by 0.1–20 µM LY2109761, a selective TGF-β receptor type 1/2 (TGF-βR1/2) kinase inhibitor. While 0.1, 1, and 5 μM LY2109761 was not cytotoxic, 10 and 20 μM LY2109761 was cytotoxic (Appendix A, see Appendix A). We then investigated pSmad-2 and sex-determining region Y-box 2 (Sox2) levels using Western blotting. Consistent with the CCK-8 assay results, the data showed that 0.1–10 μM LY2109761 dose-dependently downregulated pSmad-2 and Sox2 levels (Appendix A).

Notably, we confirmed that TGF-βR2 level in Ishikawa cells after incubation with 25 µM DEHP was significantly downregulated 0.7-fold and 0.6-fold (all *p* < 0.001) due to 1 and 5 µM LY2109761 treatment, respectively, as assessed by Western blot assay (Figure 2B). Moreover, the pSmad-2 level in Ishikawa cells after incubation with 25 µM DEHP was also potently reduced 0.5-fold and 0.4-fold (all *p* < 0.001) due to 1 and 5 µM LY2109761 treatment, respectively (Figure 2B). These results indicate that the stimulatory effect of DEHP on TGF-βR2 and pSmad-2 expression was abolished by LY2109761 treatment, indicating that the TGF-β signaling pathway was activated by DEHP exposure in human endometrial epithelial cells.

Next, we determined whether DEHP affects cell proliferation through the TGF-β signaling pathway. Based on the CCK-8 assay results, we found that the proliferation rate of Ishikawa cells following incubation with 25 µM DEHP was enhanced 1.2-fold (*p* < 0.01), whereas DEHP-induced cell proliferation was completely abrogated by LY2109761-mediated TGF-β/Smad signaling inhibition (Figure 2C). This finding was consistent with the immunocytochemical characteristics observed through Ki-67 staining (Figure 2D). In all, these results confirmed that DEHP triggers cell proliferation through the TGF-β/Smad signaling pathway in human endometrial epithelial cells.

### 2.3. DEHP Triggers Inflammatory and Immunological Responses through the TGF-β Signaling Pathway in Human EEECs

We performed qRT-PCR analysis to detect the expression levels of key genes associated with inflammatory and immunological responses in VK2/E6E7 cells after exposure to 25 µM DEHP for 48 h to elucidate whether DEHP exposure of human EEECs produces inflammatory mediators, which play a crucial role in endometriosis pathophysiology. Human tumor necrosis factor-alpha (TNF-α), interleukin-1β (IL-1β), IL-6, and interferon-γ (IFN-γ) are typical proinflammatory cytokines that mediate inflammatory and immune responses in an autocrine/paracrine/endocrine manner. The gene expression levels of *IL-1β*, *IL-6*, *TNF-α*, and *IFN-γ* were increased 3.7-fold (*p* < 0.05), 2.6-fold (*p* < 0.001), 3.1-fold (*p* < 0.001), and 1.6-fold (*p* < 0.001), respectively, compared with those in the control (Figure 3A).

Human IL-8 (C–X–C motif chemokine ligand 8, CXCL-8), monocyte chemoattractant protein-1 (MCP-1; C–C motif chemokine ligand 2, CCL2), and regulated upon activation, normal T-cell expressed and secreted (RANTES; C–C motif chemokine ligand 5, CCL5) are central proinflammatory chemokines which serve as important mediators of inflammation and immunological processes in several inflammatory and immune diseases. *IL-8*, *MCP-1*, and *RANTES* gene expression levels were augmented 2.9-fold (*p* < 0.01), 3.8-fold (*p* < 0.001), and 3.1-fold (*p* < 0.01), respectively, than those in the control (Figure 3B).

Cell adhesion molecules such as intercellular cell adhesion molecule-1 (ICAM-1), vascular cell adhesion molecule-1 (VCAM-1), and CD44 are critical for cell–cell and cell–extracellular matrix (ECM) interactions, including immune cell trafficking, and are intimately involved in many pathophysiological steps of immune-inflammatory diseases. The gene expression levels of *ICAM-1*, *VCAM-1*, and *CD44* were enhanced 1.6-fold (*p* < 0.01), 3.8-fold (*p* < 0.01), and 1.5-fold (*p* < 0.001), respectively, than those in the control (Figure 3C).

Cyclooxygenase-2 (Cox-2), which catalyzes prostaglandin formation from arachidonic acid, is induced in various pathological processes, particularly inflammation. Matrix metalloproteinases (MMPs), particularly MMP-2 (gelatinase A) and MMP-9 (gelatinase B), belong to a family of neutral proteolytic enzymes and act as vital regulators of the migration and recruitment of leukocytes and other cells during various inflammatory and immune responses. *Cox-2*, *MMP-2*, and *MMP-9* gene expression levels were elevated 2.6-fold (*p* < 0.001), 1.5-fold (*p* < 0.05), and 2.2-fold (*p* < 0.001), respectively, than those in the control (Figure 3D). Similarly, the expression levels of all three genes were significantly increased in Ishikawa cells after 25 µM DEHP treatment for 48 h (Figure 4).

Remarkably, it was also verified using qRT-PCR assay that DEHP-induced upregulation of *IL-1β*, *IL-6*, *TNF-α*, *IFN-γ*, *IL-8*, *MCP-1*, *RANTES*, *ICAM-1*, *VCAM-1*, *CD44*, *Cox-2*, *MMP-2*, and *MMP-9* expression was robustly abolished by LY2109761 (Figure 5).

### 2.4. DEHP Mediates EMT through TGF-β Signaling Pathway in Human Endometrial Epithelial Cells

EMT is an evolutionarily conserved cellular process where the expression of genes responsible for maintaining the epithelial phenotype is replaced by the expression of those responsible for maintaining the mesenchymal phenotype [23]. TGF-β has been implicated as a major EMT mediator, which represents an important pathophysiological mechanism underlying endometriosis [25]. First, to confirm whether DEHP influences EMT acquisition in Ishikawa cells, the levels of EMT mediators were assessed by Western blotting after treatment with different DEHP concentrations for 24 h. Therefore, we found that incubation with 5, 10, and 25 µM DEHP significantly increased the expression of major EMT mediators such as neural-cadherin (N-cadherin) by 1.7-fold (*p* < 0.05), 2.0-fold (*p* < 0.001), and 2.1-fold (*p* < 0.01), respectively; vimentin by 1.5-fold (*p* < 0.01), 1.5-fold (*p* < 0.001), and 1.6-fold (*p* < 0.001), respectively; Slug by 1.8-fold, 1.8-fold, and 2.4-fold (all *p* < 0.05), respectively; Snail by 1.4-fold (*p* < 0.001), 1.4-fold (*p* < 0.01), and 1.4-fold (*p* < 0.01), respectively; Twist by 2.3-fold, 2.8-fold, and 3.2-fold (all *p* < 0.001), respectively; and zinc finger E-box binding homeobox 1 (Zeb1) by 1.7-fold (*p* < 0.01), 2.1-fold (*p* < 0.001), and 2.1-fold (*p* < 0.001), respectively (Figure 6A). In contrast, incubation with 10 and 25 µM DEHP significantly reduced epithelial-cadherin (E-cadherin) expression by 0.8-fold (*p* < 0.01) and 0.5-fold (*p* < 0.001), respectively (Figure 6A).

Subsequently, we found that LY2109761 restored the DEHP-induced E-cadherin expression decrease, as assessed by flow cytometry and qRT-PCR (Figure 6B,C). Moreover, DEHP-induced upregulation of *N-cadherin*, *vimentin*, *Slug*, *Snail*, *Twist*, *Zeb1*, and *Zeb2* expression, detected by qRT-PCR assay, was repressed by LY2109761 (Figure 6D). Similarly, Slug and Snail protein expression levels were significantly increased in End1/E6E7 and VK2/E6E7 human endometriotic epithelial cells through the TGF-β signaling pathway after 25 µM DEHP treatment for 48 h (Figure 7).

### 2.5. TGF-β Signaling Is Essential in DEHP-Induced Human Endometrial Epithelial Cell Migration

Cell migration, a fundamental biological phenomenon that establishes and maintains the proper organization of multicellular organisms, is involved in many biological processes, including embryological development, tissue formation and homeostasis, and wound healing; however, cell movement can be deregulated and contributes to many pathological processes such as inflammation, cancer progression, and metastasis [30]. EMT is a critical pathophysiological process in epithelial cell migration/invasion and plays a vital role in endometriosis pathogenesis [22]. As our data showed that the expression of EMT-associated molecules was remarkably upregulated in human endometrial epithelial cells after DEHP exposure, we expected that cells activated by DEHP treatment would display elevated migratory activity compared to the untreated control cells. To assess this hypothesis, a wound healing assay was performed on human endometrial epithelial cells. The rate of wound closure at 24 h in the control and 25 µM DEHP-treated cells was 34.5% and 55.1%, respectively, whereas the DEHP-induced cell motility was robustly suppressed 1.1-fold and 0.6-fold (all *p* < 0.001) due to 1 and 5 µM LY2109761 pretreatment, respectively (Figure 8). Collectively, these data provide evidence that DEHP facilitates human endometrial epithelial cell migration and promotes their metastatic potential through the TGF-β/Smad signaling pathway.

### 2.6. DEHP Augments Human Endometrial Epithelial Cell Stemness Traits through TGF-β Signaling Pathway

We determined whether DEHP regulates the acquisition of stemness properties in human endometrial epithelial cells and assessed the expression of stemness mediators by Western blotting after treatment with different DEHP concentrations for 24 h. We found that incubation with 5, 10, and 25 µM DEHP significantly increased the expression of key stemness mediators such as Sox2 by 1.4-fold, 1.8-fold, and 2.0-fold (all *p* < 0.001), respectively; octamer-binding protein 4 (Oct4) by 1.2-fold, 1.2-fold, and 1.2-fold (all *p* < 0.01), respectively; Nanog homeobox (Nanog) by 1.1-fold, 1.3-fold (*p* < 0.05), and 1.6-fold (*p* < 0.001), respectively; and Krüppel-like factor 4 (Klf4) by 1.0-fold, 1.0-fold, and 1.2-fold (*p* < 0.01), respectively (Figure 9A). We also found that DEHP-induced Sox2, Oct4, Nanog, and Klf4 expression upregulation, detected by Western blotting, was abolished by LY2109761 (Figure 9B).

### 2.7. Formation and Growth of Human Endometrial Epithelial Cell Spheroids

For spheroid formation and growth, Figure 10A shows phase contrast microscopy images of 2D- and 3D-cultured human endometrial epithelial cells over time. These cells started to form multiple spheroids on the third day and the number of spheroids gradually increased over time (Figure 10A). The average spheroid diameters measured on days 1, 3, 5, 8, 10, and 14 were 6.8, 32.6, 57.4, 101.2, 148.6, and 257.3 µm, respectively (Figure 10B). We found that spheroids were similar in size and shape over time (Figure 10A,B). On day 10, most spheroids were approximately 150 µm in diameter. This diameter is comparable to the general size for spheroid-based drug screening, indicating that a favorable milieu for the growth of human endometrial epithelial cells was provided by the 3D cell spheroids, which can be applied for the study of the pathogenesis and development of diagnostic and therapeutic strategies for endometriosis.

### 2.8. Proliferation and Viability of Human Endometrial Epithelial Cells Are Facilitated after 3D Culture

A CCK-8 cell proliferation assay was used to quantify the ability of 3D culture to facilitate cell proliferation. Human endometrial epithelial cells were successfully propagated in the hydrogels. The cell number in 2D cultures was significantly greater than that in 3D cultures during the first 5 days (Figure 10C). However, after 8 days, the number of cells in 3D cultures exceeded those in 2D cultures (Figure 10C). The human endometrial epithelial cells showed 1.3-fold, 2.3-fold, and 3.3-fold (all *p* < 0.001) higher proliferation in 3D cultures than those in 2D cultures on days 8, 10, and 14, respectively (Figure 10C). On day 14, 2D cultures displayed 2.0-fold (*p* < 0.001) increase in cell number than that on day 1, whereas 3D cultures not only exhibited a dramatic enhancement in cell number with a 11.3-fold increase (*p* < 0.001) relative to that on day 1 (Figure 10C), but also revealed that most cells were alive as determined by live–dead staining (Figure 10D).

### 2.9. Stemness Marker Expression of Human Endometrial Epithelial Cells Is Upregulated after 3D Culture

We examined the expression of Sox2, a crucial regulator of stemness and pluripotency in stem cells, to confirm whether human endometrial epithelial cells can acquire stemness traits in our 3D cell culture model. Sox2 expression in the 3D-cultured cells for 5, 7, 10, and 14 days showed time-dependent robust increases of 1.1-fold (*p* < 0.01), 1.4-fold (*p* < 0.001), 1.6-fold (*p* < 0.001), and 2.1-fold (*p* < 0.001), respectively, compared with that in the 2D-cultured cells (Figure 10E).

### 2.10. Migratory Potential of Human Endometrial Epithelial Cells Is Elevated after DEHP Treatment in 3D Culture

Human endometrial epithelial cells grown in 3D models, which can more accurately recapitulate the complexity of in vivo biological systems, may exhibit a greater physiological responsiveness of migrating cells to DEHP than those grown in traditional 2D monolayers. Thus, we speculated that cells isolated from 3D multicellular spheroids would exhibit enhanced migratory potential than those isolated from 2D culture. Likewise, we also expected that 3D-cultured cells activated by DEHP treatment would display elevated migratory activity than the 2D-cultured control cells.

To assess these hypotheses, a wound healing assay was performed on human endometrial epithelial cells. The rate of wound closure in 2D and 3D cultures after 5 µM DEHP treatment for 24 h was 27.4% and 31.3% (all *p* < 0.001), respectively, demonstrating that the 3D niche facilitated the migration of endometrial cells more favorably than that of the 2D-cultured cells after DEHP exposure (Figure 11A). We also found that human endometrial epithelial cells generated by 3D culture in hydrogels displayed expedited migratory capability compared to those cultured in 2D. We speculated that the cells from multicellular 3D spheroids would display an enhanced mesenchymal phenotype compared with those cultured in 2D. Next, to explore the molecular mechanism by which the 3D niche and DEHP treatment regulate endometriotic cell behavior and disease progression in endometriosis, we evaluated the expression of key EMT-related molecules essential for endometriotic cell aggressiveness. Notably, DEHP treatment markedly upregulated the expression of pivotal EMT-related markers, such as N-cadherin and vimentin, in 3D cultures compared with that in 2D cultures (Figure 11B).

### 2.11. DEHP Augments EMT and Stemness through TGF-β Signaling Pathway in 3D-Cultured Endometrial Epithelial Cells

To confirm that DEHP and the 3D culture environment influence the acquisition of stemness and EMT properties in human endometrial epithelial cells through the TGF-β signaling pathway, 2D- and 3D-cultured cells were treated with 5 μM DEHP and 10 μM LY2109761. First, pSmad-2 expression was significantly enhanced 1.4-fold (*p* < 0.001) and 1.5-fold (*p* < 0.01) in 3D culture compared with that of 2D culture in both control and DEHP-treated groups, respectively, whereas the DEHP-induced increase in pSmad-2 expression was completely reversed by LY2109761 in the 3D culture (Figure 12A). In particular, pSmad-2 level was higher in the DEHP-treated 3D culture than that in the DEHP-untreated 3D control or the DEHP-treated 2D culture.

Second, the expression of the vital EMT regulators N-cadherin and Snail was markedly increased 1.1-fold and 7.7-fold (all *p* < 0.001), respectively, in 3D culture compared with that in 2D culture (Figure 12B). N-cadherin and Snail expression was also robustly elevated in 3D-cultured cells after DEHP treatment (Figure 12B). The DEHP-induced N-cadherin and Snail expression enhancement was abrogated by LY2109761 in the 3D culture (Figure 12B). Importantly, N-cadherin and Snail levels were higher in the DEHP-treated 3D culture than in the DEHP-untreated 3D control or the DEHP-treated 2D culture.

Third, the expression of crucial transcriptional regulators of stemness, namely Sox2, Oct4, and Klf4, was significantly increased 1.1-fold, 9.8-fold, and 2.1-fold (all *p* < 0.001), respectively, in 3D culture compared with that in 2D culture. The expression of Sox2, Oct4, and Klf4 was also significantly increased in 3D-cultured cells after DEHP treatment (Figure 12C), an effect that was reversed by LY2109761 (Figure 12C). Notably, the levels of Sox2, Oct4, and Klf4 were higher in the DEHP-treated 3D culture than those in the DEHP-untreated 3D control or the DEHP-treated 2D culture.

## 3. Discussion

Endometriosis, a fairly common disease in women of reproductive age, remains an enigmatic condition that is difficult to treat. Although several studies have indicated that DEHP, the most extensively used phthalate plasticizer and a ubiquitous environmental contaminant, is conceived to be associated with endometriosis pathogenesis, the underlying molecular mechanisms remain largely unknown [6,31]. In this study, we demonstrated using 2D and 3D cell culture methods that DEHP promoted proliferation, immunoinflammatory responses and cell migration and induced EMT and stemness characteristics in several human EEECs. Moreover, these DEHP-induced cellular responses, with features that indicate endometriosis, occurred via the TGF-β/Smad signaling pathway. This provides novel and critical insights into the pathogenetic role played by DEHP and its mechanisms of action in the development of endometriosis.

To our knowledge, this is the first report demonstrating that DEHP stimulates human EEEC proliferation via TGF-β/Smad signaling activation. Huang et al. [5] have reported that up to 200 μM DEHP does not affect human endometrial cell viability, whereas Kim et al. [32] have showed that DEHP exposure increases the number of human endometrial cells. Therefore, our data revealed a strong stimulatory effect of DEHP on human EEEC proliferation.

Several studies suggest that inflammation plays a vital role in the pathogenesis of endometriosis and is an important cause of endometriosis-mediated infertility [33]. Proinflammatory cytokines such as TNF-α, IL-1β, IL-6, and IFN-γ initiate and amplify inflammatory and immune responses by signaling the recruitment of additional proinflammatory mediators and immune cells to the site of injury and have been implicated in endometriosis development and progression [34]. Further, increased levels of TNF-α, IL-1β, IL-6, and IFN-γ have been detected in the serum, peritoneal fluid, and endometrium of women with endometriosis [35]. In addition to mononuclear cells, endometrial epithelial cells contribute to the production of these cytokines in endometriosis [36].

Likewise, previous studies have demonstrated that the levels of proinflammatory chemokines, particularly IL-8, MCP-1, and RANTES, which are synthesized by endometriotic cells, are elevated in the peritoneal fluid of women with endometriosis and commensurate with the disease stage [37]. Furthermore, the levels of cell adhesion molecules such as ICAM-1, VCAM-1, and CD44, which modulate cell–matrix and cell–cell attachments, are increased in the serum, peritoneal fluid, and endometrium of patients with endometriosis. These molecules are involved in regulating endometrial cell proliferation, activation, motility, chemotaxis, adhesion, morphogenesis, and implantation [38,39].

Moreover, Cox-2 is overexpressed in endometriotic epithelial and stromal cells and its principal metabolic product, prostaglandin E_2_, a hallmark of most inflammatory lesions, is a biologically active signaling lipid mediator known as an eicosanoid, which regulates many pathophysiological processes in the development and exacerbation of endometriosis, including cell proliferation, anti-apoptosis, migration, invasion, immune escape, angiogenesis, pelvic pain, dysmenorrhea, and infertility [40]. Many studies have also shown that MMPs such as MMP-2 and MMP-9 play a pivotal role in the ectopic implantation of endometrial cells via ECM degradation, particularly of the basement membrane [41]. Therefore, these inflammatory mediators are regarded as potential biomarkers of endometriosis [39].

DEHP significantly enhances *IL-1β*, *IL-6*, *IL-8*, and *TNF-α* mRNA expression in human macrophages [42], and DEHP exposure induces IL-6 and TNF-α production, thereby promoting liver inflammation, necrosis, and fibrosis in rats [8]. Inflammatory cell infiltration was also observed in DEHP-induced thyroid damage in vivo [43]. Kim [44] has observed that ICAM-1, VCAM-1, MMP-2, and MMP-9 levels were markedly increased in vascular smooth muscle cells after DEHP treatment.

However, limited information is available regarding the effect of DEHP on the expression of these inflammatory mediators in endometriosis. Only one study reported that DEHP stimulates *IL-1β*, *IL-8*, *ICAM-1*, *Cox-2*, and *MMP-2* expression in endometrial and endometriotic cells [32]. Notably, in this study, we identified a prominent DEHP-triggered molecular signature typifying endometrial inflammation, revealing high *IL-1β*, *IL-6*, *TNF-α*, *IFN-γ*, *IL-8*, *MCP-1*, *RANTES*, *ICAM-1*, *VCAM-1*, *CD44*, *Cox-2*, *MMP-2*, and *MMP-9* levels in human EEECs following DEHP exposure. In particular, from the perspective of the type and range of inflammatory markers assessed by gene expression analysis, which encompasses most major inflammatory mediators, it is plausible that our study is the first to provide definitive evidence to confirm the assumption that DEHP is capable of provoking inflammatory responses in endometrial and endometriotic cells. In addition, we demonstrated that the TGF-β signaling pathway may be an important regulator of DEHP-induced expression of inflammatory molecules in the developmental processes of endometriosis. Thus, our data provide a comprehensive and explicit insight into the role of DEHP and its molecular mechanism in triggering the immunoinflammatory response of human EEECs.

EMT is involved in the formation of endometriotic lesions by overexpressing Twist, Snail, and Slug and reducing the expression of E-cadherin [45]. Although a single study sought to determine the effects of DEHP on the expression of EMT markers, they failed to find any significant alterations in E-cadherin and vimentin levels in endometrial and endometriotic cells after DEHP exposure [32]. In contrast, we demonstrated that DEHP induces EMT in human EEECs, and furthermore, our 3D cell culture method provided a more favorable cell growth milieu than the conventional 2D cell culture, resulting in a more sensitive response to DEHP in the 3D-cultured cells. The discrepancy between these two studies may be attributed to the detection sensitivity of the analysis methods. In this context, it is considered that EMT is induced by the DEHP-stimulated inflammatory environment in human EEECs [46]. Furthermore, we demonstrated that the TGF-β/Smad signaling pathway mediates DEHP-triggered EMT induction in the pathophysiological processes of endometriosis. Likewise, TGF-β/Smad signaling also enhances EMT in cancer progression [47].

The TGF-β pathway plays a central role in inducing EMT in several different tissue types [25,26,48]. TGF-β binds to TGF-βR1 and TGF-βR2 complexes, thus phosphorylating Smad-2 and Smad-3 to form Smad complexes, which are shuttled into the nucleus as transcription factors to regulate the expression of genes implicated in mesenchymal state acquisition, invasion, motility, cell growth, proliferation, angiogenesis, and apoptosis [24]. Additionally, TGF-β signaling plays a pivotal role in tissue repair, homeostasis, wound healing, and fibrosis [49]. Thus, it is cardinally significant that our data provide a novel insight into the role of DEHP in the etiology and pathophysiology of endometriosis by promoting EMT response in human EEECs through the TGF-β/Smad signaling pathway.

The role of endometrial stem cells in the pathogenesis of endometriosis has gained considerable attention in recent years. Growing evidence supports that endometrial stem cells, as clonogenic cells in endometrial lesions, may be responsible for endometriosis development and progression, and in fact, this basis represents the concept of the stem cell origin theory of endometriosis [50]. Endometrial stem cells present in menstrual blood may be disseminated in the pelvic cavity by retrograde menstruation, leading to the generation of endometriosis [51]. Thus, endometrial stem cells arising from the uterine endometrium or bone marrow may be involved in endometriosis pathophysiology [2,50]. The progenitor/stem cell populations in the endometrial basalis layer in women of reproductive age are important because they have an extraordinary proliferative potential to regenerate the endometrium at about 5–7 mm within one week of every menstrual cycle, as they can generate approximately 6 × 10^11^ cells from a single cell [52]. Thus, endometrial stem cell activity cannot be paralleled to that of other organs and is vital for endometrial function.

Several studies have revealed that the activation of EMT in epithelial cells induces the acquisition of stem-cell properties in cancer, indicating a link between EMT and stemness [53]. N-cadherin was proposed as the first specific surface marker for human endometrial progenitor/stem cells, which are located at the base of endometrial glands adjacent to the myometrium, and their capacity for self-renewal and differentiation was verified [54]. N-cadherin positive epithelial cells were detected in 50% of peritoneal and deep infiltrating endometriotic lesions [54].

Stemness and pluripotency factors, such as Sox2, Nanog, Oct4, and Klf4, are master transcriptional regulators that maintain the pluripotent embryonic stem cell phenotype. The expression of stemness-related genes, namely, *Oct4*, *Sox2*, and *Nanog*, was markedly increased in the ectopic endometrium of patients with endometriosis, and thus supports a role for stem cells in endometriosis pathogenesis [55]. Several studies have focused on the expression of stemness-related factors in endometriotic lesions, suggesting that self-renewal rates and stem cell fates are dysregulated in endometriosis, thereby leading to altered stem cell behavior [56]. Thus, it is speculated that certain stimuli that can change the stemness properties of endometrial cells may trigger the development and progression of endometriosis. As a salient feature of this study, we demonstrated that DEHP upregulated the expression of N-cadherin and key stemness markers, including Sox2, Nanog, Oct4, and Klf4, in human endometrial epithelial cells, indicating that DEHP alters their stemness status, thereby contributing to the pathogenesis of endometriosis. Therefore, stemness may be enhanced by DEHP-induced EMT in human endometrial epithelial cells. Moreover, we demonstrated that the TGF-β/Smad signaling pathway mediates DEHP-triggered stemness augmentation in human endometrial epithelial cells, leading to the development and progression of endometriosis. Likewise, TGF-β signals stimulate EMT and promote endometrial cell migration in endometriotic lesions through Oct4 expression upregulation [57].

Our hypothesis on the role of DEHP in driving stemness in human endometrial epithelial cells is also strongly supported by the recent observation that DEHP affects the activity of other stem cell types. Our data are consistent with that of a previous study showing that DEHP profoundly stimulates neural stem cell proliferation through *Sox2* overexpression [58]. Concurrently, Chen et al. [59] have found that the levels of stemness-related proteins, including Oct4, Sox2, and Nanog, were upregulated in DEHP-treated colon cancer cells. Accordingly, it is conceivable that DEHP exposure promotes stemness, rendering the cells motile, clonogenic, phenotypically plastic, and resistant to immunosurveillance and cell death. However, adverse effects of DEHP have also been reported in certain types of stem cells, including hematopoietic and embryonic stem cells [60,61]. These controversial findings may be attributed to potentially different cellular responses to DEHP depending on the cell type and nature, genetic and epigenetic traits, and microenvironmental characteristics. In all, it is exceedingly significant that our data provide compelling evidence proving that DEHP boosts the stemness potential of human endometrial epithelial cells through the TGF-β/Smad signaling pathway, thereby predisposing to the development of endometriosis.

Three-dimensional cell culture is increasingly recognized as a pivotal tool for recapitulating a variety of in vivo biological phenomena [62]. Although several studies have developed 3D in vitro cell culture models for endometriosis to decipher its complexity, no definitive nor ideal model has yet been established [27,28,29]. Interestingly, we observed that our 3D culture model is more suitable for the growth and functional maintenance of human EEECs in many aspects than 2D monolayer culture; this assertion was demonstrated by the fact that cells grown in 3D culture exhibited augmented survival, proliferation, migration, and gene and protein expression than those grown in 2D culture. In line with the fact that plasma DEHP concentrations in patients with endometriosis range from 1.5 to 6.2 μM [5], our data suggest that our 3D culture model evaluated with 5 μM DEHP, which is a clinically relevant concentration, more favorably reflects biological responses than traditional monolayer models. Hence, unlike 2D cell culture, our 3D human EEEC culture model has great potential in facilitating the understanding of endometriosis etiology and pathobiology as well as developing diagnostic, therapeutic, and preventive strategies for endometriosis.

## 4. Materials and Methods

### 4.1. Cell Culture and Reagents

End1/E6E7 (ATCC CRL-2615) and VK2/E6E7 (ATCC CRL-2616) were purchased from the American Type Culture Collection (ATCC; Manassas, VA, USA) and cultured in serum-free medium (Gibco, Waltham, MA, USA) with 0.1 ng/mL human epidermal growth factor, 0.05 mg/mL bovine pituitary extract, and 0.4 mM CaCl_2_. The Ishikawa cell line (European Collection of Cell Cultures, Salisbury, UK) was cultured in Roswell Park Memorial Institute medium (Hyclone, Logan, UT, USA) supplemented with 10% fetal bovine serum (FBS; Welgene, Gyeongsan-si, Korea) and 1% penicillin–streptomycin (Gibco). DEHP and dimethyl sulfoxide (DMSO) were purchased from Sigma-Aldrich (St. Louis, MO, USA). The selective TGF-β receptor type 1/2 (TGF-βR1/2) kinase inhibitor LY2109761 was obtained from Selleck (Houston, TX, USA).

### 4.2. Hydrogel Synthesis for 3D Cell Culture

A biomimetic hydrogel was prepared for the 3D culture of Ishikawa cells, as previously reported [63,64]. The marine collagen, agarose, and alginate solutions were combined with 1 × 10^5^ cells/mL. For gelation, the hydrogel solutions were vortexed and incubated at 4 °C for 5 min.

### 4.3. Cell Proliferation Assay

For 2D and 3D cell cultures, 1 × 10^4^ Ishikawa cells were incubated in each well of a 96-well plate. Cell viability was measured using a Cell Counting Kit-8 (CCK-8) assay (Dongin, Seoul, Korea). The optical density at 450 nm was measured using a microplate reader (Tecan, Männedorf, Switzerland). The cell morphology was also assessed using a phase contrast microscope (IX70; Olympus, Tokyo, Japan).

### 4.4. Spheroid Growth Assay

To observe multicellular Ishikawa spheroid formation and growth in the hydrogels, the spheroid sizes were measured at the desired time points using a phase contrast microscope (Olympus). At least 15 spheroids on each hydrogel were photographed, and their diameters were measured. The diameter of a spheroid was defined as the average diameter measured at two-degree intervals joining two outline points and passing through the centroid. The spheroid diameter was quantified and analyzed using ImageJ software version 1.52a (National Institute of Health, Bethesda, MD, USA).

### 4.5. Evaluation of Immunofluorescence Using Confocal Microscopy

The cells were cultured in 8-well glass slides (SPL Life Sciences, Pocheon, Korea), which were used for immunofluorescence analysis to evaluate Ki-67 protein expression in proliferating cells. Monolayer cells were fixed with 4% paraformaldehyde in phosphate-buffered saline (PBS) for 10 min, followed by permeabilization with 0.1% Triton X-100 in PBS for 10 min. The cells were incubated with Ki-67 monoclonal antibody (1:100; clone SolA15; eBioscience, San Diego, CA, USA) at 4 °C overnight. After washing in PBS, the samples were incubated for 1 h at room temperature (RT) with DyLight 594-conjugated goat anti-mouse IgG secondary antibody (1:100; Bethyl Laboratories, Montgomery, TX, USA). Immunofluorescence was observed using a laser confocal microscope (LSM900; Carl Zeiss, Jena, Germany).

### 4.6. Wound Healing Assay

Ishikawa cells (5 × 10^5^ cells/well) were seeded in 6-well plates. When the cells reached complete confluence, the medium was replaced with a starvation medium containing 0.5% FBS. Scratch wounds were created using a scratcher (SPL Life Sciences) in the cells of each well. After rinsing with PBS, scratch closure was monitored and imaged using a phase contrast microscope (Olympus).

### 4.7. RNA Isolation and cDNA Synthesis

Total RNA from cells was isolated using TRIzol reagent (Favorgen Biotech Corp, Pingtung, Taiwan). RNA quality and quantity were determined using a NanoDrop 2000 spectrophotometer (Thermo Scientific, Waltham, MA, USA). First-strand cDNA was synthesized with 1 µg total RNA from each sample using a HiSenScript™ RH(−) RTase cDNA Synthesis Kit (iNtRON Biotechnology, Sungnam, Korea). The reaction mixture was incubated at 45 °C for 60 min and then heated to 85 °C for 10 min to stop the reaction.

### 4.8. Quantitative Real-Time PCR (qRT-PCR)

The Bio-Rad CFX Connect Real-Time system (Hercules, CA, USA) and the DNA-binding dye SYBR Green I with the SsoAdvanced Universal SYBR Green Supermix (Bio-Rad) were used to perform qRT-PCR. The specific primers used for qRT-PCR are listed in Table 1. All samples were amplified in triplicate. Gene expression levels were calculated using the 2^−ΔΔCt^ method and normalized to GAPDH expression levels. The expression of the control sample was set at 1, and the relative expression of other samples was calculated accordingly.

### 4.9. Western Blot Analysis

Total protein was extracted by lysing cells in radio-immunoprecipitation assay (RIPA) lysis buffer (GenDEPOT, Barker, TX, USA) containing a protease inhibitor cocktail (GenDEPOT) for 30 min on ice. The protein concentration was determined using a bicinchoninic acid protein assay (Sigma-Aldrich). Equal amounts of protein were loaded onto a 12% (*v/v*) sodium dodecyl sulfate-polyacrylamide gel for electrophoresis. The separated proteins were blotted onto a polyvinylidene fluoride membrane (Amersham Biosciences, Piscataway, NJ, USA). Membranes were first blocked with 3% BSA in Tris-buffered saline with 0.1% Tween-20 at RT for 1 h and then incubated overnight at 4 °C with primary polyclonal antibodies (1:1000 dilution; Table 2). The blots were then incubated with peroxidase-conjugated anti-mouse and anti-rabbit secondary antibodies (7076 and 7074, respectively, Cell Signaling Technology, Danvers, MA, USA) diluted at 1:10,000 for 1 h at RT. The target proteins were visualized using an enhanced chemiluminescence kit (Amersham Biosciences), imaged using an Amersham Imager 680 (Amersham Biosciences) and quantified with ImageJ software. The density of each band was normalized to that of β-actin.

### 4.10. Flow Cytometry

Ishikawa cells (1 × 10^6^ cells/tube) were incubated with a primary antibody against epithelial-cadherin (E-cadherin; 1:100; Abcam, Cambridge, UK) for 1 h at RT. The cells were incubated with an anti-mouse Alexa Fluor 488-conjugated secondary antibody (1:100 dilution) for 30 min at RT. Fluorescence-activated cell sorting analysis was performed using a FACS Canto-II flow cytometer (BD Biosciences, San Jose, CA, USA). Flow cytometry data were analyzed using FlowJo 10.3.0 (Tree Star; Ashland, OR, USA).

### 4.11. Statistical Analysis

All quantitative results were expressed as the mean ± standard deviation (SD) of at least three independent experiments. Comparisons between two groups were analyzed using Student’s *t*-test. Values of *p* < 0.05 were considered statistically significant.

## 5. Conclusions

Our study demonstrated that 5–25 μM DEHP exposure exhibited distinctive pathophysiological characteristics of endometriosis in 2D and 3D cultures of human EEECs: (1) enhanced cell proliferation; (2) facilitated cell migration; (3) upregulated expression of crucial proinflammatory cytokines, such as IL-1β, IL-6, TNF-α, and IFN-γ; (4) augmented expression of critical proinflammatory chemokines, including IL-8, MCP-1, and RANTES; (5) increased expression of cell adhesion molecules, such as ICAM-1, VCAM-1, and CD44, which are key players in inflammation and immune response; (6) boosted the expression of vital inflammatory mediators, namely, Cox-2, MMP-2, and MMP-9; (7) induction of EMT, as evidenced by overexpressed EMT-driving transcription factors such as Zeb1, Zeb2, Snail, Slug, and Twist as well as reduced E-cadherin expression; and (8) acquisition of stem cell-like features, as exemplified by the heightened expression of N-cadherin and the pivotal stemness markers Sox2, Nanog, Oct4, and Klf4. Remarkably, we found that these effects of DEHP are mediated through TGF-β/Smad signaling pathway activation. Therefore, our data imply that DEHP is a critical causative agent of endometriosis that enhances cell proliferation and migration, triggers immunoinflammatory responses, activates EMT, and induces the stem cell phenotype in human EEECs through the TGF-β/Smad signaling pathway. Our 3D human EEEC culture model was more favorable than a conventional 2D cell culture method not only for assessing cell activities such as proliferation and migration but also for detecting gene and protein expression. Furthermore, the 3D endometriotic cell model can be useful and promising to study the biology of endometriosis in vitro as well as to develop new therapeutic measures.

## Figures and Tables

**Figure 1 ijms-23-03938-f001:**
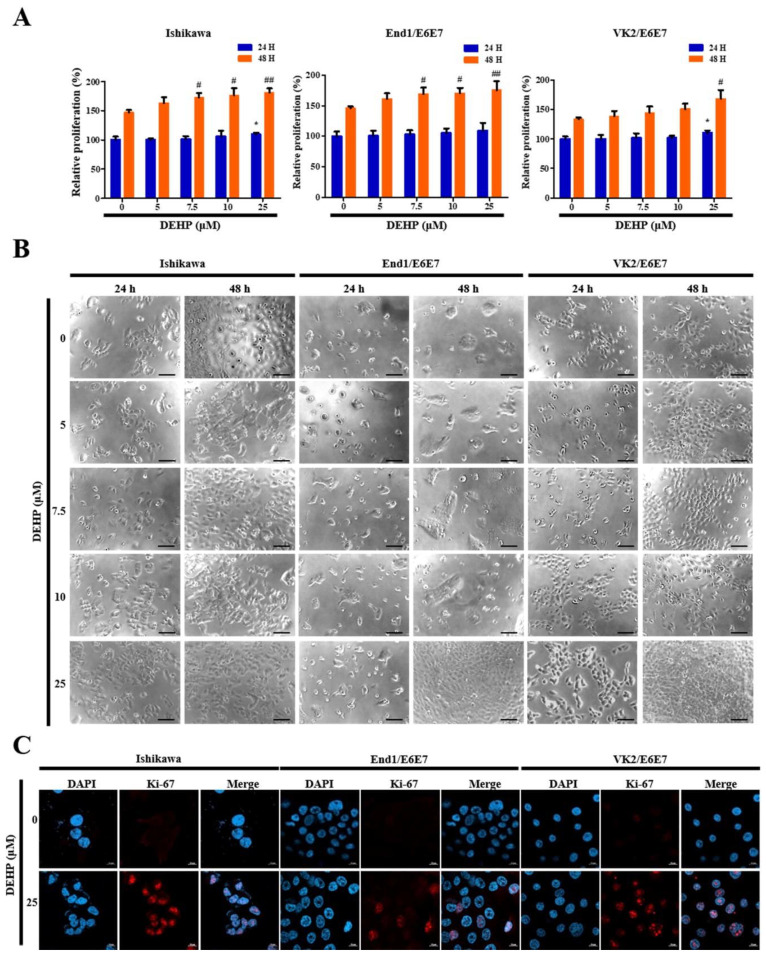
Di-(2-ethylhexyl) phthalate (DEHP) enhances human endometrial epithelial cell proliferation through transforming growth factor (TGF)-β signaling pathway. (**A**) Cell proliferation increased dose-dependently by 2, 5, 7.5, 10, and 25 µM DEHP treatment for 24 and 48 h; (**B**) photomicrographs (100×) under a phase contrast microscope showing cell morphology in the control and DEHP-treated cells for 24 and 48 h; (**C**) representative fluorescence micrographs of Ki-67 staining (blue: DAPI; red: Ki-67). Cells treated with 25 μM DEHP for 48 h show a strong nuclear immunostaining. All data are expressed as relative values against their respective control group. Data represent the mean ± standard deviation of three independent experiments. * *p* < 0.05 (vs. 24 h control), # *p* < 0.05 and ## *p* < 0.01 (vs. 48 h control). Scale bars = 10 μm.

**Figure 2 ijms-23-03938-f002:**
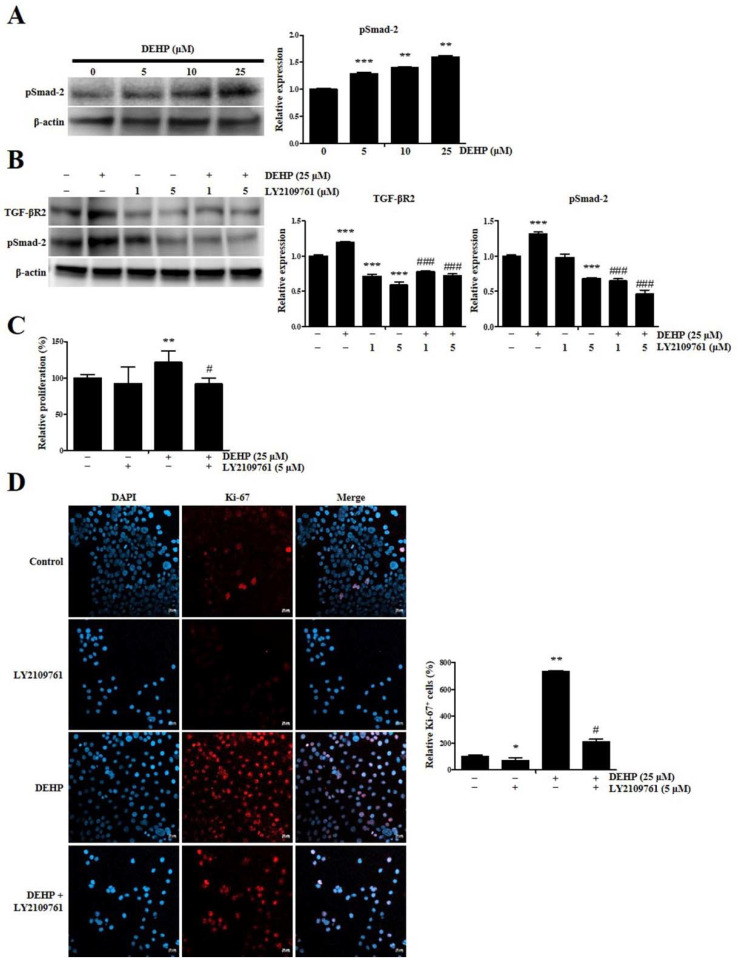
Di-(2-ethylhexyl) phthalate (DEHP) enhances human endometrial epithelial cell proliferation through transforming growth factor (TGF)-β signaling pathway. (**A**) pSMAD-2 level detected by Western blotting after treatment with 5, 10, and 25 µM DEHP for 24 h. β-actin was used as the internal loading control for data normalization. (**B**) LY2109761 (0.1 to 20 μM) displays cellular toxicity dose-dependently. (**C**) LY2109761 (0.1 to 10 μM) represses pSMAD-2 and SOX2 protein levels, as detected using Western blot analysis. β-actin was used as the internal loading control for data normalization. (**D**) TGF-βR2 and pSMAD-2 protein levels and quantitative analysis of their protein levels after treatment with 25 µM DEHP and LY2109761. β-actin was used as the internal loading control for data normalization. DEHP—induced cell proliferation is completely abrogated by LY2109761−mediated TGF−β/Smad signaling inhibition. Representative fluorescence micrographs of K-67 staining (blue: DAPI; red: K-67). All data are expressed as relative values against their respective control group. Data represent the mean ± standard deviation of three independent experiments. * *p* < 0.05, ** *p* < 0.01, and *** *p* < 0.001 (vs. control); # *p* < 0.05 and ### *p* < 0.001 (vs. DEHP treatment). Scale bars = 20 μm.

**Figure 3 ijms-23-03938-f003:**
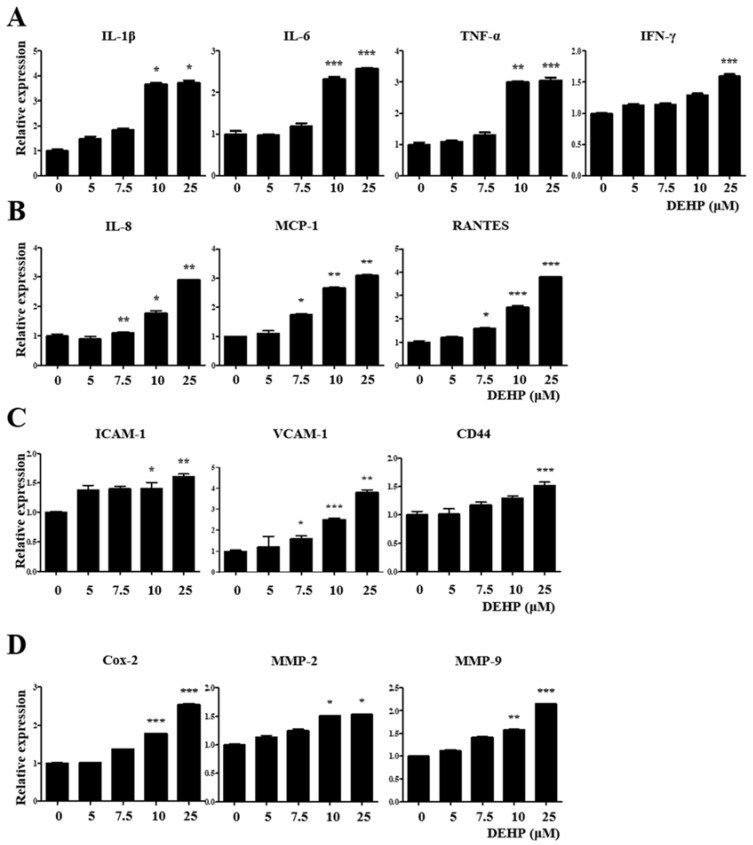
Di-(2-ethylhexyl) phthalate (DEHP) triggers the expression of inflammatory and immunological mediators in human endometriotic epithelial cells as detected by quantitative reverse transcription-PCR (qRT-PCR). mRNA expression levels of (**A**) proinflammatory cytokines *IL-1β*, *IL-6*, *TNF-α*, and *IFN-γ*; (**B**) proinflammatory chemokines *IL-8*, *MCP-1*, and *RANTES*; (**C**) cell adhesion molecules *ICAM-1*, *VCAM-1*, and *CD44*; (**D**) *COX-2*, *MMP-2*, and *MMP-9*. Glyceraldehyde-3-phosphate dehydrogenase (*GAPDH*) was used as the housekeeping gene. All data are expressed as relative values against their respective control group. Data represent the means ± standard deviation of three independent experiments. * *p* < 0.05, ** *p* < 0.01, and *** *p* < 0.001 (vs. control).

**Figure 4 ijms-23-03938-f004:**
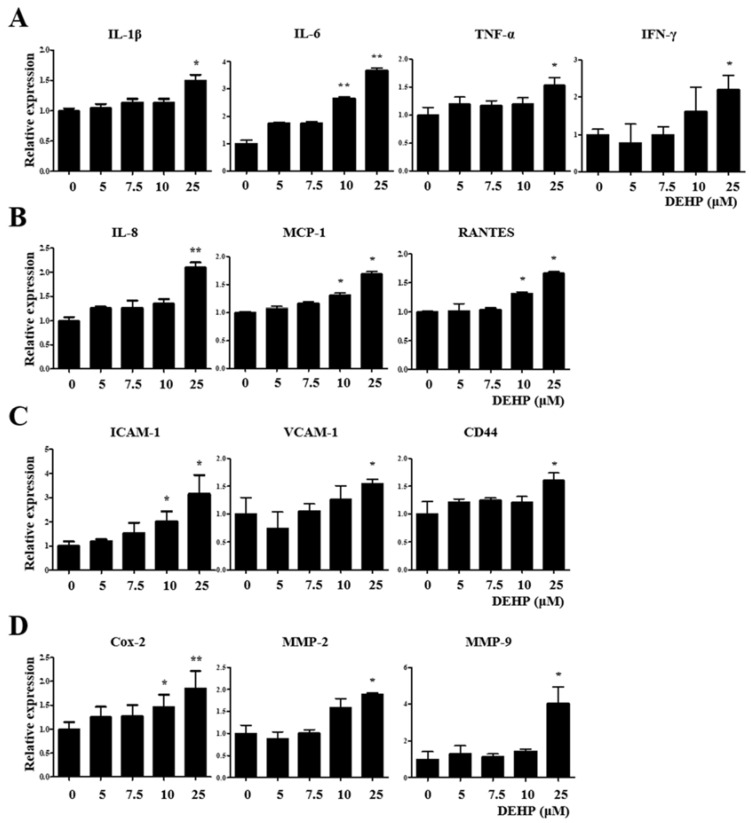
Di-(2-ethylhexyl) phthalate (DEHP) triggers the expression of inflammatory and immunological mediators in human endometrial epithelial cells, as detected by quantitative reverse transcription-PCR (qRT-PCR). mRNA expression levels of (**A**) proinflammatory cytokines *IL-1β*, *IL-6*, *TNF-α*, and *IFN-γ*; (**B**) proinflammatory chemokines *IL-8*, *MCP-1*, and *RANTES*; (**C**) cell adhesion molecules *ICAM-1*, *VCAM-1*, and *CD44*; (**D**) *COX-2*, *MMP-2*, and *MMP-9*. Glyceraldehyde-3-phosphate dehydrogenase (*GAPDH*) was used as the housekeeping gene. All data are expressed as relative values against their respective control group. Data represent the means ± standard deviation of three independent experiments. * *p* < 0.05 and ** *p* < 0.01(vs. control).

**Figure 5 ijms-23-03938-f005:**
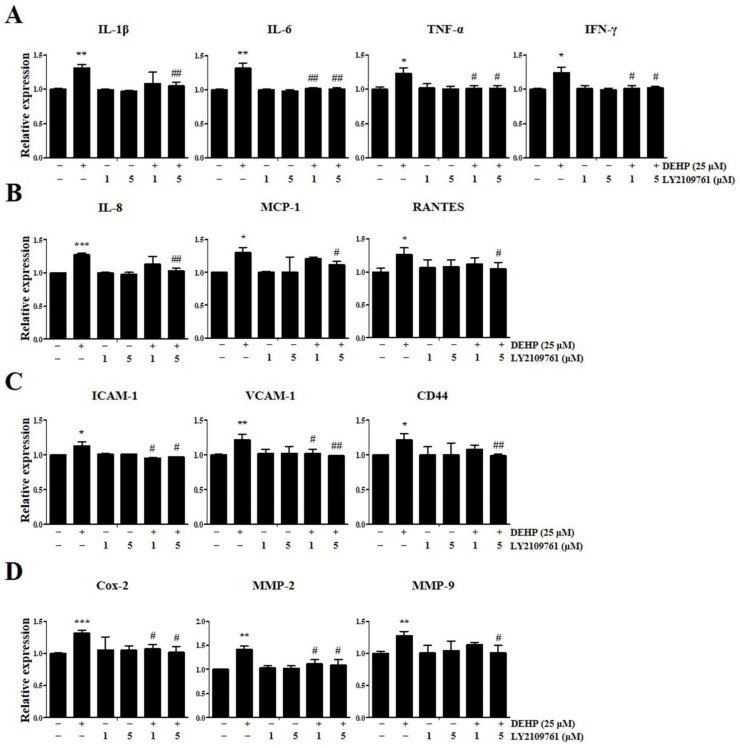
Di-(2-ethylhexyl) phthalate (DEHP) triggers immunoinflammatory responses through transforming growth factor (TGF)-β signaling pathway in human endometriotic epithelial cells as detected using quantitative reverse transcription-PCR (qRT-PCR). mRNA expression levels of (**A**) proinflammatory cytokines *IL-1β*, *IL-6*, *TNF-α*, and *IFN-γ*; (**B**) proinflammatory chemokines *IL-8*, *MCP-1*, and *RANTES*; (**C**) cell adhesion molecules *ICAM-1*, *VCAM-1*, and *CD44*; (**D**) inflammatory mediators *COX-2*, *MMP-2*, and *MMP-9*. Glyceraldehyde-3-phosphate dehydrogenase (*GAPDH*) was used as the housekeeping gene. All data are expressed as relative values against their respective control group. Data represent the mean ± standard deviation of three independent experiments. * *p* < 0.05, ** *p* < 0.01, and *** *p* < 0.001 (vs. control); # *p* < 0.05 and *## p <* 0.01 (vs. DEHP treatment).

**Figure 6 ijms-23-03938-f006:**
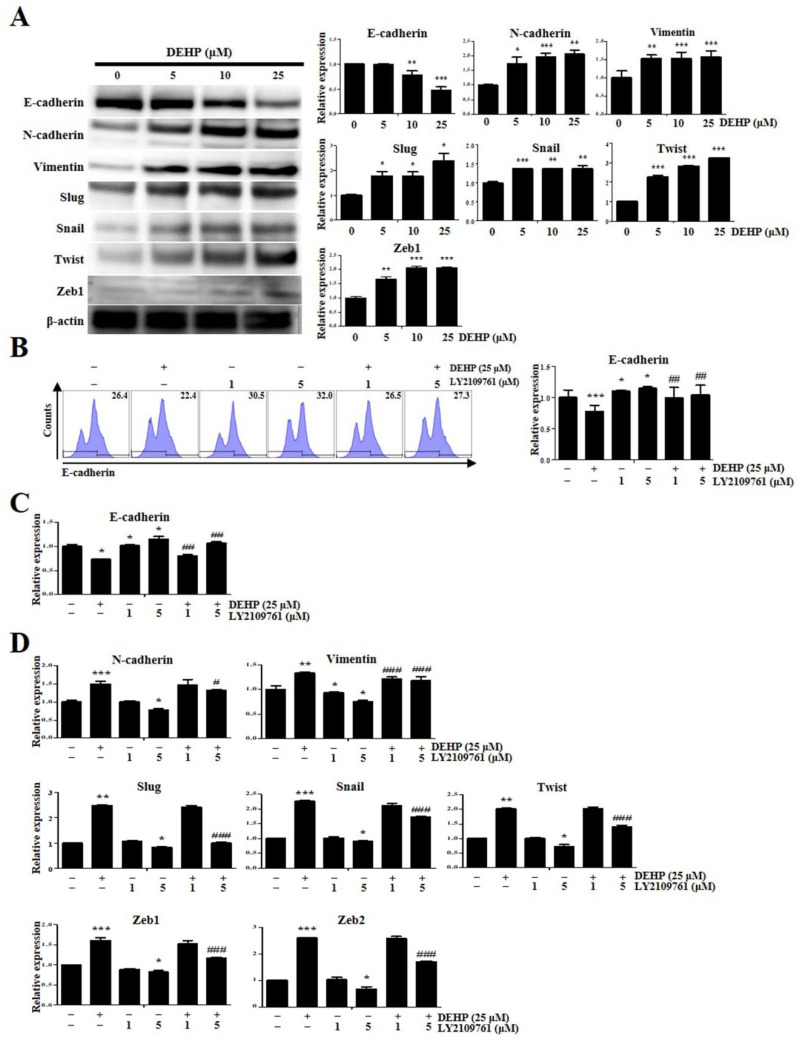
Di-(2-ethylhexyl) phthalate (DEHP) mediates epithelial–mesenchymal transition through transforming growth factor (TGF)-β pathway in human endometrial epithelial cells. (**A**) E-cadherin, N-cadherin, vimentin, SLUG, SNAIL, TWIST, and ZEB1 expression after DEHP treatment. LY2109761 restores the decreased E−cadherin expression as assessed using (**B**) flow cytometry and (**C**) quantitative reverse transcription-PCR (qRT-PCR). (**D**) qRT-PCR analysis showing alleviated *N-cadherin*, *vimentin*, *SLUG*, *SNAIL*, *TWIST*, *ZEB1*, and *ZEB2* mRNA levels via TGF-β signaling pathway inhibition. All data are expressed as relative values against their respective control group. Data represent the mean ± standard deviation of three independent experiments. * *p* < 0.05, ** *p* < 0.01, and *** *p* < 0.001 (vs. control); # *p* < 0.05, ## *p* < 0.01, and ### *p* < 0.001 (vs. DEHP treatment).

**Figure 7 ijms-23-03938-f007:**
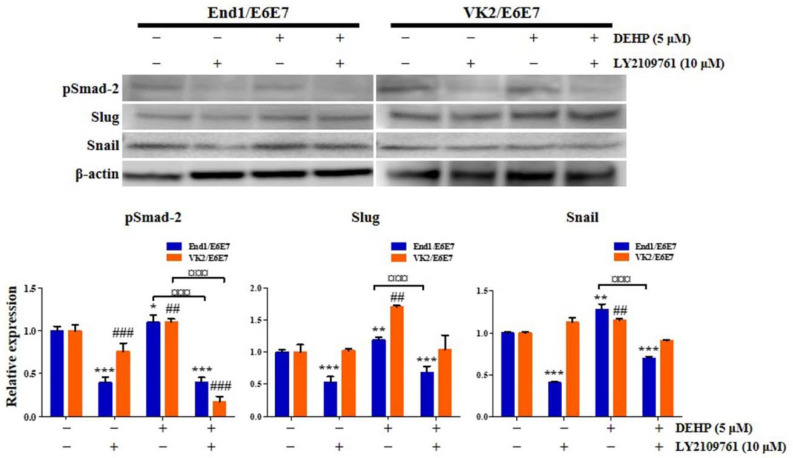
Di-(2-ethylhexyl) phthalate (DEHP) mediates epithelial-mesenchymal transition (EMT) through TGF-β signaling pathway in human endometriotic epithelial cells. pSMAD-2, SLUG, and SNAIL levels detected by Western blotting. β-actin was used as the internal loading control for Western blot analysis data normalization. All data are expressed as relative values against their respective control group. Data represent the mean ± standard deviation of three independent experiments. * *p* < 0.05, ** *p* < 0.01 and *** *p* < 0.001 (vs. control); ## *p* < 0.01 and ### *p* < 0.001 (vs. control); ¤¤¤ *p* < 0.001.

**Figure 8 ijms-23-03938-f008:**
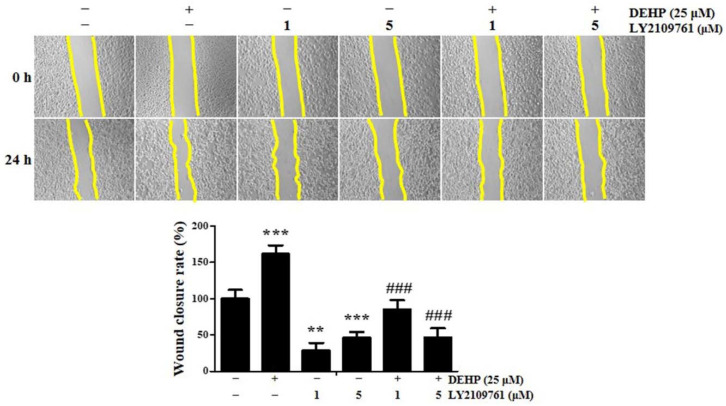
Di-(2-ethylhexyl) phthalate (DEHP)—induced elevated migratory potential of human endometrial epithelial cells. Representative photomicrographs (100×) of Ishikawa cells migrating to the cell-free space under a phase contrast microscope. Images were captured at 0 and 24 h. The distances between the two edges are scaled for three positions at different times. All data are expressed as relative values against their respective control group. Data represent the mean ± standard deviation of three independent experiments. ** *p* < 0.01 and *** *p* < 0.001 (vs. control); ### *p* < 0.001 (vs. DEHP treatment).

**Figure 9 ijms-23-03938-f009:**
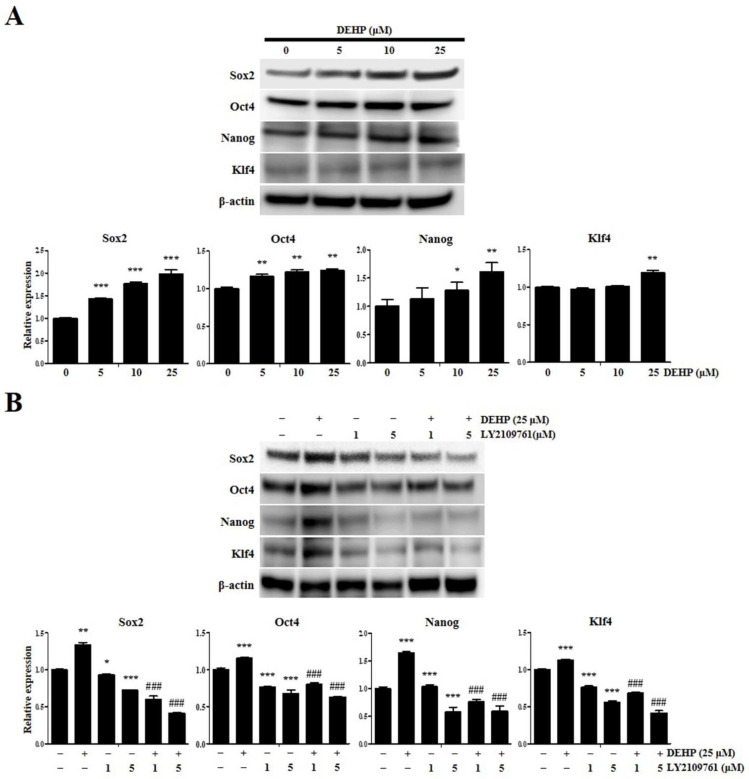
Di-(2-ethylhexyl) phthalate (DEHP) augments human endometrial epithelial cell stemness traits through transforming growth factor (TGF)-β signaling pathway. (**A**) SOX2, OCT4, NANOG, and KLF4 levels detected using Western blotting after 5, 10, and 25 µM DEHP treatment for 24 h. (**B**) Western blot analysis showing the alleviated SOX2, OCT4, NANOG, and KLF4 protein expression levels via inhibiting the TGF-β signaling pathway. β-actin was used as the internal loading control for data normalization. All data are expressed as relative values against their respective control group. Data represent the mean ± standard deviation of three independent experiments. * *p* < 0.05, ** *p* < 0.01, and *** *p* < 0.001 (vs. control); ### *p* < 0.001 (vs. DEHP treatment).

**Figure 10 ijms-23-03938-f010:**
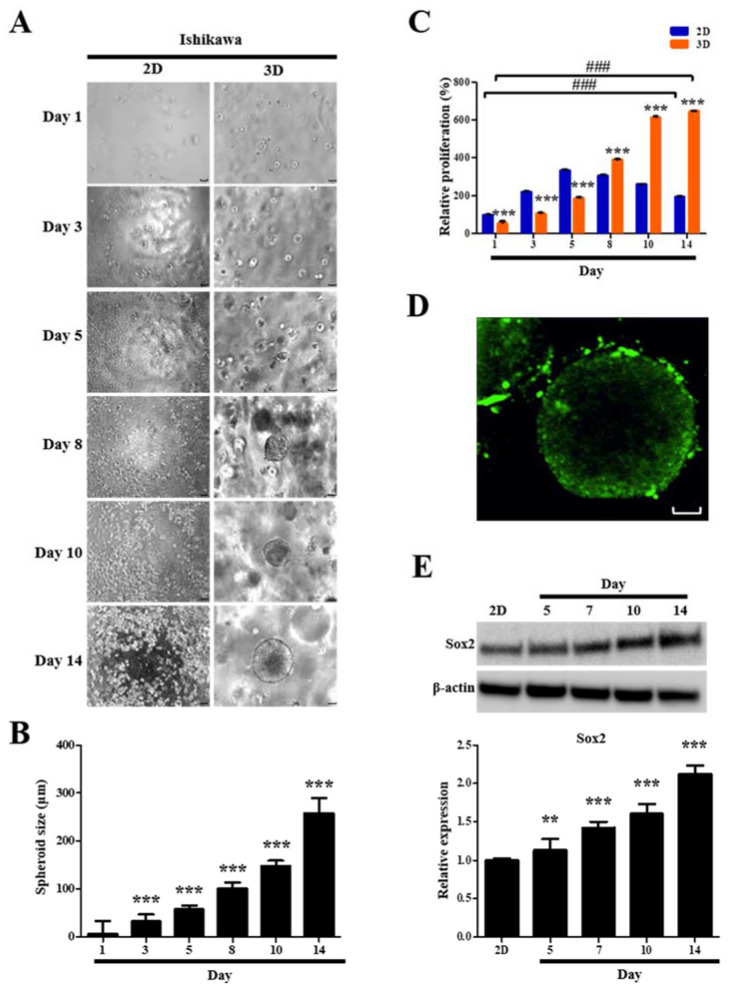
Formation and growth of human endometrial epithelial cell spheroids in standard plastic tissue culture plates and hydrogels. (**A**) Photomicrograph (100×) under a phase contrast microscope showing images of 2D- and 3D-cultured cells with time; (**B**) diameters of cell spheroids depicted in a bar graph; (**C**) CCK-8 cell proliferation assay of 2D- and 3D-cultured cells; (**D**) representative fluorescence microscopic image (400×) of 3D-cultured cells on day 14 as determined using live–dead cell viability assay (Green, calcein: live cells; red, EthD: dead cells). (**E**) SOX2 protein level increased time-dependently in 3D-cultured cells, as detected by Western blotting. β-actin was used as the internal loading control for Western blot analysis data normalization. All data are expressed as relative values against their respective control group. Data represent the mean ± standard deviation of three independent experiments. ** *p* < 0.01 and *** *p* < 0.001; ### *p* < 0.001. Scale bars = 50 μm.

**Figure 11 ijms-23-03938-f011:**
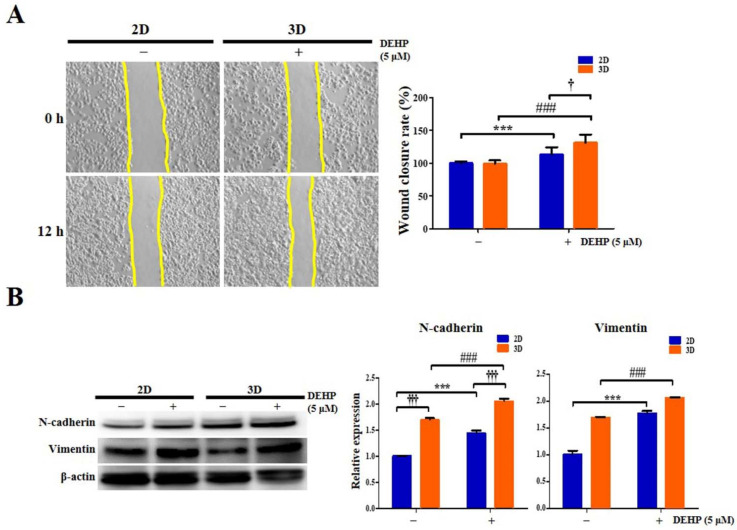
Epithelial-mesenchymal transition (EMT) characteristics of human endometrial epithelial cells are augmented after di-(2-ethylhexyl) phthalate (DEHP) treatment in 3D culture as detected using Western blot analysis. (**A**) Representative photomicrographs (100×) of 2D- and 3D-cultured Ishikawa cells migrating to the cell−free space under a phase contrast microscope. Images were captured at 0 and 12 h. The distances between the two edges are scaled for three positions at different times; (**B**) N-cadherin and vimentin protein expression levels. β-actin was used as the internal loading control for data normalization. All data are expressed as relative values against their respective control group. Data represent the mean ± standard deviation of three independent experiments. *** *p* < 0.001; ### *p* < 0.001; † *p* < 0.05 and ††† *p* < 0.001.

**Figure 12 ijms-23-03938-f012:**
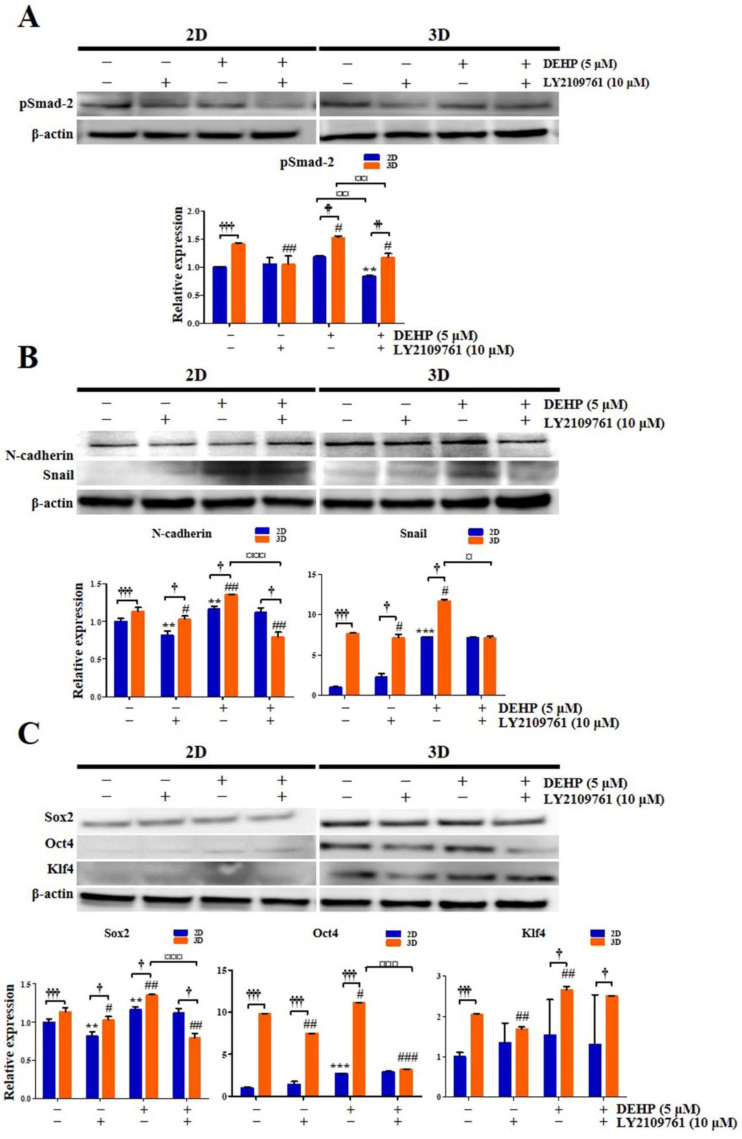
Stemness characteristics of human endometrial epithelial cells are augmented after di-(2-ethylhexyl) phthalate (DEHP) treatment in 3D culture as detected using Western blot analysis. (**A**) pSMAD-2; (**B**) N-cadherin and SNAIL; (**C**) SOX2, OCT4, and KLF4 protein expression levels. β-actin was used as the internal loading control for data normalization. All data are expressed as relative values against their respective control group. Data represent the mean ± standard deviation of three independent experiments. ** *p* < 0.01 and *** *p* < 0.001; # *p* < 0.05, ## *p* < 0.01, and ### *p* < 0.001; † *p* < 0.05, †† *p* < 0.01 and ††† *p* < 0.001; ¤ *p* < 0.05, ¤¤ *p* < 0.01 and ¤¤¤ *p* < 0.001.

**Table 1 ijms-23-03938-t001:** qRT-PCR primer names and their sequences.

Gene	Forward (5ʹ-3ʹ)	Reverse (5ʹ-3ʹ)
*CD44*	CTGCCGCTTTGCAGGTGTA	CATTGTGGGCAAGGTGCTATT
*COX-2*	GCCTGAATGTGCCATAAGACTGAC	AAACCCACAGTGCTTGACACAGA
*E-cadherin*	ATTTTTCCCTCGACACCCGAT	TCCCAGGCGTAGACCAAGA
*ICAM-1*	CCGGAAGGTGTATGAACTGA	GGCAGCGTAGGGTAAGGTT
*IFN-γ*	TGGCTTTTCAGCTCTGCATC	CCGCTACATCTGAATGACCTG
*IL-1β*	CCTGTCCTGCGTGTTGAAAGA	GGGAACTGGGCAGACTCAAA
*IL-6*	GGTACATCCTCGACGGCATCT	GTGCCTCTTTGCTGCTTTCAC
*IL-8*	GCATAAAGACATACTCCAAACC	ACTTCTCCACAACCCTCTG
*Klf4*	CAGCTTCACCTATCCGATCCG	GACTCCCTGCCATAGAGGAGG
*MCP-1*	CCGAGAGGCTGAGACTAACC	CTTTCATGCTGGAGGCGAGA
*MMP-2*	TGACGGTAAGGACGGACTC	ATACTTCACACGGACCACTTG
*MMP-9*	TTGACAGCGACAAGAAGTGG	GCCATTCACGTCGTCCTTAT
*NANOG*	CAAAGCAGGAGTCCACTGAG	TAAGGGCATCCACTTCACAG
*N-cadherin*	AGCCAACCTTAACTGAGGAGT	GGCAAGTTGATTGGAGGGATG
*OCT4*	CTTGAATCCCGAATGGAAAGGG	GTGTATATCCCAGGGTGATCCTC
*RANTES*	TCTGCGCTCCTGCATCTG	AGTGGGCGGGCAATGTAG
*SNAIL*	ACTGCAACAAGGAATACCTCAG	GCACTGGTACTTCTTGACATCTG
*SLUG*	TGTGACAAGGAATATGTGAGCC	TGTGACAAGGAATATGTGAGCC
*SOX2*	TACAGCATGTCCTACTCGCAG	GAGGAAGAGGTAACCACAGGG
*TNF-α*	CCCAGGGACCTCTCTCTAATC	ATGGGCTACAGGCTTGTCACT
*Vimentin*	CAAAGCAGGAGTCCACTGAG	TAAGGGCATCCACTTCACAG
*VCAM-1*	ACACCTCCCCCAAGAATACAG	GCTCATCCTCAACACCCACAG
*ZEB1*	TTACACCTTTGCATACAGAACCC	TTTACGATTACACCCAGACTGC
*ZEB2*	GCGATGGTCATGCAGTCAG	CAGGTGGCAGGTCATTTTCTT
*GAPDH*	GGAGAAGGCTGGGGCTCAT	TGATGGCATGGACTGTGGTC

**Table 2 ijms-23-03938-t002:** Antibodies used for western blotting.

Antibody	Source	Antibody Type	Size (kDa)
E-cadherin	Abcam	Mouse monoclonal	110
KLF4	Abcam	Rabbit polyoclonal	55
NANOG	Abcam	Rabbit polyclonal	35
N-cadherin	Abcam	Mouse monoclonal	100
OCT4	Abcam	Rabbit polyclonal	43
pSMAD-2	Cell Signaling Technology	Rabbit monoclonal	60
SLUG	Cell Signaling Technology	Rabbit monoclonal	30
SNAIL	Cell Signaling Technology	Rabbit monoclonal	29
SOX2	Abcam	Rabbit polyclonal	34
TGF-βR2	Abcam	Rabbit polyclonal	65
TWIST	Abcam	Mouse monoclonal	21
Vimentin	Abcam	Rabbit polyclonal	54
ZEB1	Cell Signaling Technology	Rabbit monoclonal	200
β-actin	Santa Cruz Biotechnology	Mouse monoclonal	42

## Data Availability

Not applicable.

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
