# Peer review of "Di-(2-ethylhexyl) Phthalate Triggers Proliferation, Migration, Stemness, and Epithelial–Mesenchymal Transition in Human Endometrial and Endometriotic Epithelial Cells via the Transforming Growth Factor-β/Smad Signaling Pathway"

_ijms, 2022, doi:10.3390/ijms23073938_

Round 1

Reviewer 1 Report

This is a very interesting piece if work where the authors report DEHP induced proliferation, migration, stemness and EMT in endometrial cells. The authors successfully demonstrated that the activation of TGF-beta/SMAD pathway is responsible for these changes. Overall the paper is scientifically sound and well presented. There are some comments which I believe the authors should address:

  1. Figure 1B- 24 and 48 h data. Not a good representation of what has been shown in figure 1A. The panel in 1B does not show any increased proliferation after 48 hour treatment. The figures should be replaced with one that is better representative.
  2. It is good practice to initially use more than 1 drug inhibitor at various concentrations (nm-um). (for information)
  3. Similarly it is important to use 2-3 house keeping genes/proteins in blots and gene expression experiments to ensure validity of the data (for information) 
  4. Very long sentence (52-62). It would be good to mention the type of products and cite a couple of references. There is no need to list each product.

    Some sentences such as 144-146 need to be rewritten. The structure is poor.

Reviewer 2 Report

I read with great interest the manuscript, which falls within the aim of this Journal. In my honest opinion, the topic is interesting enough to attract the readers’ attention.

Well done. A really well written paper. I only have a small suggestion:

- Introduction part, from line 75 to 81, references are missing.

- It would be appreciable if you can add some theories on etiopathogenesis, citing those about the altered intestinal absorption that can cause transposition of intestinal microbiota and the inflammation of the peritoneal fluid (PMID: 32401078), highlighting the relationships between endometriosis and the host microbiome (PMID: 33851803)
